# FAST ADVERSARIAL TRAINING FOR SEMI-SUPERVISED LEARNING

## ABSTRACT

In semi-supervised learning, *Bad GAN* approach is one of the most attractive method due to the intuitional simplicity and powerful performances. *Bad GAN* learns a classifier with bad samples distributed on complement of the support of the input data. But *Bad GAN* needs additional architectures, a generator and a density estimation model, which involves huge computation and memory consumption cost. *VAT* is another good semi-supervised learning algorithm, which utilizes unlabeled data to improve the invariance of the classifier with respect to perturbation of inputs. In this study, we propose a new method by combining the ideas of *Bad GAN* and *VAT*. The proposed method generates bad samples of high-quality by use of the adversarial training used in *VAT*. We give theoretical explanations why the adversarial training is good at both generating bad samples and semi-supervised learning. An advantage of the proposed method is to achieve the competitive performances with much fewer computations. We demonstrate advantages our method by various experiments with well known benchmark image datasets.

## 1 INTRODUCTION

Deep learning has accomplished unprecedented success due to the development of deep architectures, learning techniques and hardwares (Krizhevsky et al., 2012; Ioffe & Szegedy, 2015; Szegedy et al., 2015; Hinton et al., 2012; Kingma & Ba, 2014). However, deep learning has also suffered from collecting large amount of labeled data which requires both cost and time. Thus it becomes important to develop semi-supervised methodologies that learn a classifier (or discriminator) by using small labeled data and large unlabeled data.

Various semi-supervised learning methods have been proposed for deep learning. Weston et al. (2012) employs a manifold embedding technique using the pre-constructed graph of unlabeled data and Rasmus et al. (2015) uses a specially designed auto-encoder to extract essential features for classification. Variational auto encoder (Kingma & Welling, 2013) is also used in the context of semi-supervised learning by maximizing the variational lower bound of both labeled and unlabeled data (Kingma et al., 2014; Maaløe et al., 2016).

Recently, semi-supervised learning based on generative adversarial networks (*GAN*, Goodfellow et al. (2014a)) has received much attention. For $K$-class classification problems, Salimans et al. (2016) solves the $(K + 1)$-class classification problem where the additional $(K + 1)$th class consists of synthetic images made by a generator of the *GAN* learned by unlabeled data. Dai et al. (2017) notices that not a good generator but a bad generator which generates synthetic images much different from observed images is crucial for the success of semi-supervised learning. Dai et al. (2017) gives theoretical justifications of using a bad generator and develops a semi-supervised learning algorithm called *Bad GAN* which achieves the state-of-the-art performances over multiple benchmark datasets.

However, *Bad GAN* has several limitations. It needs two additional deep architectures - bad generator and pre-trained density estimation model besides the one for the classifier. Learning these multiple deep architectures requires huge computation and memory consumption. In particular, the *PixelCNN++* (Salimans et al., 2017) is used for the pre-trained density estimation model which needs very large computational resources.

Another difficulty in *Bad GAN* is that it requires a two-step learning procedure - the first step is to learn the *PixelCNN++* model and the second step is to learn the classifier and the bad generator. The optimal learning of the first step may not be optimal for the second step and hence the regularization of the both steps would need special techniques.

In this study, we propose a new semi-supervised learning method which competes well with other state-of-the-art semi-supervised learning algorithms and yet needs much smaller amount of computational resources. In particular, the proposed method employs only one deep architecture and hence the corresponding learning phase is much easier and faster.

Our proposed method is motivated by close investigation of *VAT* (Virtual Adversarial Training) method (Miyato et al., 2015; 2017). *VAT* tries to find a deep classifier which has a good prediction accuracy on training data and at the same time is less sensitive to data perturbation toward the adversarial direction. Here, the adversarial direction for a given datum is the direction to which the probabilities of each class change most. In Section 3, we prove that the perturbed data toward their adversarial directions can serve as 'good' bad samples. By using the adversarial directions for both measuring the invariance and generating the bad samples, the proposed method combines the advantages of *Bad GAN* and *VAT* together. Note that only a deep architecture for classification is needed to calculate the adversarial directions and thus the corresponding learning procedure is cheaper, easier and faster. We call our proposed method *FAT* (Fast Adversarial Training).

Dai et al. (2017) proves that bad samples play a role to pull the decision boundary toward the low density regions of data. In Section 5, we give a theoretical explanation that *VAT* pushes the decision boundary away from the high density regions of data. That is, *FAT* accelerates the learning procedure by using both pushing and pulling operations simultaneously. In section 6, we show that *FAT* achieves almost the state-of-the-art performances with much fewer training epochs. Especially, for the MNIST dataset, *FAT* achieves similar test accuracies to those of *Bad GAN* and *VAT* with 5 times and 7 times fewer training epochs, respectively.

This paper is organized as follows. In Section 2, we review the *Bad GAN* and *VAT* methods briefly. In Section 3, the technique to generate bad samples using the adversarial directions is described, and our proposed semi-supervised learning method is presented in Section 4. Theoretical analysis of *VAT* is given in Section 5. Results of various experiments are presented in Section 6 and conclusions follow in Section 7.

## 2 *Bad GAN* AND *VAT*

### 2.1 *Bad GAN* APPROACH

*Bad GAN* is a method that trains a good discriminator with a bad generator. This procedure trains a generator as well as a discriminator simultaneously. Let $\mathcal{D}_G(\phi)$ be generated bad samples with a bad generator $p_G(\cdot; \phi)$ parametrized by $\phi$. Here, the 'bad generator' is a deep architecture to generate samples different from observed data. Let $p^{\text{pt}}(\cdot)$ be a pre-trained density estimation model. For a given discriminator with a feature vector $v(x; \theta)$ of a given input $x$ parametrized by $\theta$, *Bad GAN* learns the bad generator by minimizing the following:

$$\mathbb{E}_{x \sim \mathcal{D}_G(\phi)} \left[ \log p^{\text{pt}}(x) \mathbb{I}(p^{\text{pt}}(x) > \tau) \right] + ||\mathbb{E}_{x \sim \mathcal{U}^{tr}} v(x; \widehat{\theta}) - \mathbb{E}_{x \sim \mathcal{D}_G(\phi)} v(x; \widehat{\theta})||^2$$

with respect to $\phi$, where $\tau > 0$ is a tuning parameter, $\mathcal{U}^{tr}$ is the unlabeled data, *and $\widehat{\theta}$ is the current* estimate of $\theta$ and $\| \cdot \|$ is the Euclidean norm.

In turn, to train the discriminator, we consider the $K$-class classification problem as the $(K + 1)$-class classification problem where the $(K + 1)$-th class is an artificial label of the bad samples generated by the bad generator. We estimate the parameter $\theta$ in the discriminator by minimizing the following:

$$-\mathbb{E}_{x,y\sim\mathcal{L}^{tr}}\left[\log p(y|x, y \leq K; \theta)\right] - \mathbb{E}_{x\sim\mathcal{U}^{tr}}\left[\log\left\{\sum_{k=1}^{K} p(k|x; \theta)\right\}\right]$$

$$-\mathbb{E}_{x\sim\mathcal{D}_G(\phi)}\left[\log p(K + 1|x; \theta)\right] - \mathbb{E}_{x\sim\mathcal{U}^{tr}}\left[\sum_{k=1}^{K} p(k|x; \theta) \log p(k|x; \theta)\right] \quad (1)$$

for given $\phi$, where $\mathcal{L}^{tr}$ is the labeled set. The second and the third terms in (1) are the cross-entropies between the unlabeled and the bad samples. The fourth term is similar the entropy of the unlabeled data which is usually helpful for semi-supervised learning (Grandvalet & Bengio, 2005). See Dai et al. (2017) for details of the objective function (1).

## 2.2 *VAT* APPROACH

*VAT* is a regularization method which is inspired by the adversarial training (Goodfellow et al., 2014b). The regularization term of *VAT* is given as:

$$\begin{aligned} L^{\text{VAT}}(\theta; \widehat{\theta}, x, \epsilon) &= D_{\text{KL}}\left(p(\cdot|x; \widehat{\theta})||p(\cdot|x + r_{\text{advr}}(x, \epsilon); \theta)\right) \\ &= -\sum_{k=1}^{K} p(k|x; \widehat{\theta}) \log p(k|x + r_{\text{advr}}(x, \epsilon); \theta) + C, \end{aligned}$$

where

$$r_{\text{advr}}(x, \epsilon) = \underset{r;||r||\leq\epsilon}{\operatorname{argmax}} D_{\text{KL}}\left(p(\cdot|x; \widehat{\theta})||p(\cdot|x + r; \widehat{\theta})\right), \quad (2)$$

$\epsilon > 0$ is a tuning parameter, $\theta$ is the parameter in the discriminator to train, $\widehat{\theta}$ is the current estimate of $\theta$ and $C$ is a constant. Combining with the cross-entropy term of the labeled data, we get the final objective function of *VAT*:

$$-\mathbb{E}_{x,y\sim\mathcal{L}^{tr}}\left[\log p(y|x; \theta)\right] + \mathbb{E}_{x\sim\mathcal{U}^{tr}}\left[L^{\text{VAT}}(\theta; \widehat{\theta}, x, \epsilon)\right]. \quad (3)$$

## 3 GENERATION OF BAD SAMPLES BY ADVERSARIAL TRAINING

The key role of bad samples in *Bad GAN* is to enforce the decision boundary to be pulled toward the low density regions of the unlabeled data. To do this, bad samples must be located at the valleys of the distribution of the unlabeled data. In this section, we propose a novel technique to generate 'good' bad samples by use of only a given classifier.

### 3.1 MOTIVATION

In this subsection, we explain why the adversarial direction is toward the decision boundary. For simplicity, we only consider the linear decision boundary. For the decision boundary made by the DNN model with ReLU activation function, see Appendix A.2.

Let us consider the 2-class linear logistic regression model parametrized by $\eta = \{w, b\}$, that is, $p(y = 1|x; \eta) = \left(1 + \exp(-b - w^{'}x)\right)^{-1}$. Note that the decision boundary is $\{x : b + w^{'}x = 0\}$, and for any given $x$, the distance between $x$ and the decision boundary is $|b + w^{'}x|/||w||$. The key

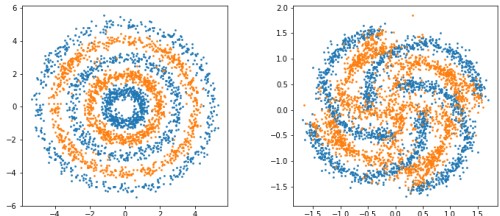

Figure 1: Demonstration of how the bad samples generated by the adversarial training are distributed. We consider two cases: 3-class classification problem (**Left**) and 4-class classification problem (**Right**). True data and bad data are coloured by blue and orange, respectively.

result is that moving $x$ toward the adversarial direction $r_{\text{advr}}(x, \epsilon)$ is equivalent to moving $x$ toward the decision boundary which is stated rigorously in the following proposition. The proof is in the appendix.

**Proposition 1** *For a sufficiently small $\epsilon > 0$, we have*

$$sign(w^{'}x + b) \cdot sign\left(w^{'}r_{advr}(x, \epsilon)\right) = -1. \tag{4}$$

Proposition 1 implies that $|b + w^{'}x|/||w|| > |b + w^{'}(x + r^{\text{advr}}(x, \epsilon)|/||w||$ unless $|b + w^{'}x| = 0$. Hence, we can treat $x + \delta r_{\text{advr}}(x, \epsilon)/||r^{\text{advr}}(x, \epsilon)||$ for appropriately choosing $\delta > 0$ as a bad sample (a sample closer to the decision boundary).

### 3.2 BAD SAMPLE GENERATION WITH GENERAL CLASSIFIER

Motivated by Proposition 1, we propose a bad sample generator as follows. Let $\delta > 0$ be fixed and $\widehat{\theta}$ be the current estimate of $\theta$. For a given input $x$ and a classifier $p(\cdot|x; \widehat{\theta})$, we calculate the adversarial direction $r_{\text{advr}}(x, \epsilon)$ for given $\epsilon$ by (2). Then, we consider $x_{\text{bad}} = x + \delta r_{\text{advr}}(x, \epsilon)/||r_{\text{advr}}(x.\epsilon)||$ as a bad sample (a sample closer to the decision boundary). We generate bad samples for all unlabeled data. In practice, we apply the same $\delta$ to all unlabeled data and choose $\delta$ based on the validation data accuracy.

In Figure 1, we illustrate how the bad samples generated by the proposed adversarial training are distributed for multi-class problem. With a good classifier, we can clearly see that most bad samples are located well in the low density regions of the data.

It may happen that a generated bad sample is not sufficiently close to the decision boundary to be a 'good' bad sample, in particular when $\delta$ is too large or too small. To avoid such a situation, we exclude $x_{\text{bad}}$ which satisfies the following condition:

$$\max_{k} p(k|x_{\text{bad}}; \widehat{\theta}) > 1 - \tau$$

for a prespecified $\tau > 0$. In our experiments, we set the optimal $\tau$ with validation data.

## 4 FAST ADVERSARIAL TRAINING WITH BAD SAMPLES

Once we generate bad samples by the adversarial training, *FAT* updates $\theta$ by minimizing the following objective function:

$$-\mathbb{E}_{x,y \sim \mathcal{L}^{tr}} \left[\log p(y|x; \theta)\right] + \mathbb{E}_{x \sim \mathcal{U}^{tr}} \left[L^{\text{true}}(\theta; x)\right] + \mathbb{E}_{x \sim \mathcal{D}^{bad}(\widehat{\theta}, \epsilon, \delta)} \left[L^{\text{fake}}(\theta, x)\right]$$
$$+\mathbb{E}_{x \sim \mathcal{U}^{tr}} \left[L^{\text{VAT}}(\theta; \widehat{\theta}, x, \epsilon)\right] \tag{5}$$

where $\mathcal{D}^{bad}(\widehat{\theta}, \epsilon, \delta)$ is the set of generated bad samples with $\widehat{\theta}, \epsilon$ and $\delta$,

$$L^{\text{true}}(\theta; x) = -\sum_{k=1}^{K} \left[ \frac{\exp(g_k(x; \theta))}{1 + \sum_{k'=1}^{K} \exp(g_{k'}(x; \theta))} \log \frac{\exp(g_k(x; \theta))}{1 + \sum_{k'=1}^{K} \exp(g_{k'}(x; \theta))} \right],$$

$$L^{\text{fake}}(\theta; x) = -\log \frac{1}{1 + \sum_{k=1}^{K} \exp(g_k(x; \theta))},$$

$g(x; \theta) \in \mathbb{R}^K$ is a pre-softmax vector of a given deep architecture and $\lambda > 0$. We treat $\epsilon$ and $\delta$ as tuning parameters to be selected based on the validation data accuracy. We minimize the objective function with one of the standard optimization algorithms such as *Adam* (Kingma & Ba, 2014) or *RMSProp* (Tieleman & Hinton, 2012).

The objective function (5) differs from the objective function (1) of *Bad GAN* in a way that the second term of (1), the cross-entropy of not being a bad sample, is deleted and the regularization term of *VAT* is added. We delete the second term of (1) because it can be easily shown that a perfect classifier of unlabeled data can be obtained from the minimizer of the objective function (1) without the second term under the conditions in Dai et al. (2017). The regularization term of *VAT* is added to improve the bad sample generator based on adversarial training. See Section 5.1 for detailed discussions.

Miyato et al. (2017) proposes the fast approximation method to calculate the adversarial direction $r_{\text{advr}}(x, \epsilon)$ by using the second-order Taylor expansion. Let us define $H(x, \widehat{\theta}) = \nabla\nabla D_{\text{KL}}\left(p(\cdot|x; \widehat{\theta})||p(\cdot|x + r; \widehat{\theta})\right)|_{r=0}$. They claim that $r_{\text{advr}}$ emerges as the first dominant eigenvector $u(x, \widehat{\theta})$ of $H(x, \widehat{\theta})$ with magnitude $\epsilon$. But there always exist two dominant eigenvectors, $\pm u(x, \widehat{\theta})$, and the sign should be selected carefully. So, we slightly modify the approximation method of Miyato et al. (2017) by

$$r_{\text{advr}}(x, \epsilon) = \underset{r \in \{u(x, \widehat{\theta}), -u(x, \widehat{\theta})\}}{\text{argmax}} D_{\text{KL}}\left(p(\cdot|x; \widehat{\theta})||p(\cdot|x + r; \widehat{\theta})\right).$$

This modification helps to improve convergence speed of the test accuracy, which will be demonstrated in the ablation experiments.

## 5 ROLE OF *VAT* FOR SEMI-SUPERVISED LEARNING

In this section, we investigate the roles of the regularization term of *VAT* in our method in detail. First, we verify that *VAT* regularization term does help to generate 'better' bad samples with a simple experiment. Furthermore, we give a theoretical insight for the role of the regularization term of *VAT* in semi-supervised learning. We will show that the regularization term of *VAT* pushes the decision boundary from the high density regions of unlabeled data. As a result, *FAT* uses pushing operations by *VAT* term as well as pulling operations by bad samples simultaneously, which makes it possible to accelerate the training process with improved performances.

### 5.1 IMPROVEMENT OF BAD SAMPLES WITH *VAT*

For generated samples by adversarial training to be 'good' bad samples, the adversarial directions should be toward the decision boundary. While this always happens for the linear model by Proposition 1, adversarial directions could be opposite to the decision boundary for deep model. To avoid such undesirable cases as much as possible, it would be helpful to smoothen the classifier with a regularization term. In this section, we explain that the regularization term of *VAT* plays such a role.

The adversarial direction obtained by maximizing the KL divergence is sensitive to local fluctuations of the class probabilities which is examplified in Figure 2. The regularization term of *VAT* is helpful

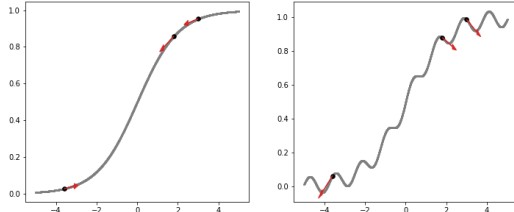

Figure 2: Examples of $P(y = 1|x)$ of smooth **(Left)** and wiggle **(Right)** cases. We plot 3 points and their adversarial directions on each case.

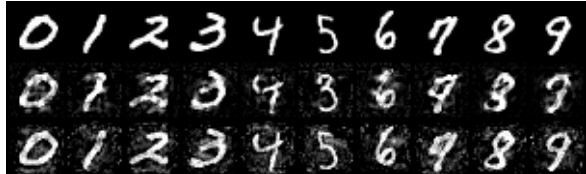

Figure 3: **(Upper)** 10 randomly sampled original MNIST dataset. **(Middle and Lower)** Bad samples obtained by the classifier learned with and without the regularization term of *VAT*.

to find a right adversarial direction which is toward the decision boundary by eliminating unnecessary local fluctuations of the class probabilities. In Figure 3, we compare bad samples generated by the adversarial training with and without the regularization term of *VAT* for the MNIST dataset. While the bad samples generated without the regularization term of *VAT* are visually similar to the given input vectors, the bad samples generated with the regularization term of *VAT* look like mixtures of two different digits and thus serve as 'better' bad samples.

### 5.2 THEORETICAL ANALYSIS OF *VAT*

Suppose that $\mathcal{X}$ is partitioned by $(K + 1)$ mutually disjoint subsets $\mathcal{X}_k$ for $k = 1, \ldots, K + 1$ such that $y(x) = k$ for all $x \in \mathcal{X}_k, k \leq K$ and $p^*(x) = 0$ for $x \in \mathcal{X}_{K+1}$, where $y(x)$ is the ground-truth label of $x$ and $p^*(x)$ is the true density of $x$. For a given feature map $u : \mathcal{X} \to \mathbb{R}^m$ and weight vectors $w_1, \ldots, w_k$, let $p(y = k|x) \propto \exp(w_k' u(x))$ for $k = 1, \ldots, K$ and $p(y = K + 1|x) \propto 1$. Let $\mathcal{L}^{tr}$ satisfy $\mathcal{X}_k \cap \mathcal{L}_{tr} \neq \emptyset$ for $k = 1, \ldots K$. Suppose that (i) $\text{argmax}_h p(y = h|x) = y$ for $x \in \mathcal{L}_{tr}$, (ii) $\text{argmax}_h p(y = h|x) \leq K$ for $x \in \cup_{k=1}^{K} \mathcal{X}_k$ and (iii) $\text{argmax}_h p(y = h|x) = K + 1$ for $x \in \mathcal{X}_{K+1}$. Under these three conditions, Dai et al. (2017) proves that $\text{argmax}_h p(y = h|x) = y(x)$ for all $x \in \cup_{k=1}^{K} \mathcal{X}_k$. That is, generating large 'good' bad samples is helpful for classifying unlabeled data correctly only with a small amount of labeled data.

A similar result can be obtained for *VAT* objective function itself under mild additional conditions. For given two subsets $\mathcal{A}_1$ and $\mathcal{A}_2$ of $\mathcal{X}$, we define $d(\mathcal{A}_1, \mathcal{A}_2) = \min_{x_1 \in \mathcal{A}_1, x_2 \in \mathcal{A}_2} d(x_1, x_2)$ for a given metric $d$. We define a tuple $(x, x')$ is $\epsilon$-*connected* if $d(x, x') < \epsilon$. A finite subset $\mathcal{A}$ of $\mathcal{X}$ is called $\epsilon$-connected iff for all $x, x' \in \mathcal{A}$, there exists a finite path $(x, x_1, ..., x_q, x')$ such that $x_j \in \mathcal{A}$ for $j = 1, \ldots, q$ and$(x, x_1), (x_1, x_2), ..., (x_{q-1}, x_q), (x_q, x')$ are all $\epsilon$-*connected*.

**Proposition 2** *Assume that there exists $\epsilon > 0$ such that **1)** $\mathcal{U}_k^{tr} = \mathcal{U}^{tr} \cap \mathcal{X}_k, k = 1, \ldots, K$ are $\epsilon$-connected, **2)** $d(\mathcal{X}_k, \mathcal{X}_{k'}) \geq 2\epsilon$ for all $k \neq k' \leq K$, and **3)** for each $k \leq K$, there exists at least one $(x, k) \in \mathcal{L}^{tr}$ such that $d(\{x\}, \mathcal{U}_k^{tr}) < \epsilon$. Suppose that there exists a classifier $f : \mathcal{X} \to \{1, ..., K\}$ such that $f(x) = y$ for all $(x, y) \in \mathcal{L}^{tr}$ and*

$$f(x) = f(x') \quad \text{for all } x' \in \mathcal{B}(x, \epsilon) \tag{6}$$

*for all $x \in \mathcal{U}^{tr}$, where $\mathcal{B}(x, \epsilon) = \{x' : d(x, x') \leq \epsilon\}$. Then, the function $f$ classifies the unlabeled set perfectly, that is:*

$$f(x) = y(x) \quad \text{for all } x \in \mathcal{U}^{tr}.$$

Condition **1)** and **2)** mean that the unlabeled data are dense enough and the supports of each class are separated sufficiently, respectively. Condition **3)** assumes that at least one labeled instance exists near the supports of each class. The main condition of Proposition 2 is (6) which essentially assumes that the classifier $f$ does not change much locally. That is, $f$ is invariant with respect to all small perturbations of an input $x$. Note that the regularization term of *VAT* is devised to improve the invariancy of the classifier and Proposition 2 explains why improving the invariancy is helpful for semi-supervised learning.

### 5.3    INTERPRETATION OF *VAT*

Let $f(x; \theta) = \text{argmax}_h p(y = h | x; \theta)$. Proposition 2 implies that it would be good to pursue a classifier which predicts the labeled data correctly and at the same time is invariant with respect to all local perturbations on the unlabeled data. For this purpose, a plausible candidate of the objective function is

$$\mathbb{E}_{(x,y)\sim\mathcal{L}^{tr}}\left[\mathbb{I}(y \neq f(x;\theta)\right] + \mathbb{E}_{x\sim\mathcal{U}^{tr}}\left[\mathbb{I}\left(f(x;\theta) \neq f(x';\theta) \text{ for } \forall x' \in \mathcal{B}(x,\epsilon)\right)\right]. \tag{7}$$

Note that a classifier $f$ which achieves 0 value of the objective function (7) satisfies the conditions of Proposition 2 and thus classifies all unlabeled data correctly.

The objective function (7) is not practically usable since neither optimizing the indicator function nor checking $f(x; \theta) \neq f(x'; \theta)$ for all $x'$ in $\mathcal{B}(x, \epsilon)$ is possible. To resolve these problems, we replace the indicator functions in (7) with the cross-entropies, and the neighborhood $\mathcal{B}(x, \epsilon)$ in the second term with the adversarial direction. By doing so, we have the following alternative objective function:

$$-\mathbb{E}_{x,y\sim\mathcal{L}^{tr}}\left[\log p(y|x;\theta)\right] - \mathbb{E}_{x\sim\mathcal{U}^{tr}}\left[\sum_{k=1}^{K} p(k|x;\theta)\log p(k|x + r_{\text{advr}}(x,\epsilon);\theta)\right]. \tag{8}$$

Finally, we replace $p(\cdot|x; \theta)$ in the second term of (8) by $p(\cdot|x; \widehat{\theta})$ to have the objective function of *VAT* (3).

The condition (6) in Proposition 2 means that the decision boundary is not located inside the support $\mathcal{X}_k$ of each class. That is, the regularization term of *VAT* prevents the decision boundary from being located at the high density regions of data or equivalently pushes the decision boundary from the high density regions of data.

## 6    EXPERIMENTS

### 6.1    PREDICTION PERFORMANCES IN SEMI-SUPERVISED LEARNING

We compare prediction performances of *FAT* over the benchmark datasets with other semi-supervised learning algorithms. We consider the most widely used datasets: MNIST (LeCun et al., 1998), SVHN (Marlin et al., 2010) and CIFAR10 (Krizhevsky & Hinton, 2009). As in done by other works, we randomly sample 100, 1000 and 4000 labeled data from the MNIST, SVHN and CIFAR10 datasets, respectively and use them as the labeled data and the rest as the unlabeled data. For fair comparison, we use the same architectures as those used in Miyato et al. (2017) for MNIST, SVHN and CIFAR10. The optimal tuning parameters $(\epsilon, \tau, \delta)$ in *FAT* are chosen based on the validation data accuracy.

Table 1: Comparison of prediction accuracies of various semi-supervised learning algorithms for the three benchmark datasets. *mod. VAT* is the modified version of *VAT* stated in Section 4.

| Method | Test acc.(%) | | |
|---|---|---|---|
| | MNIST(100) | SVHN(1000) | CIFAR10(4000) |
| *DGN* (Kingma et al., 2014) | 96.67 | 63.98 | - |
| *Ladder* (Rasmus et al., 2015) | 98.94 | - | 79.6 |
| *ALI* (Donahue et al., 2016) | - | 92.58 | 82.01 |
| *FM-GAN* (Salimans et al., 2016) | 99.07 | 91.89 | 81.37 |
| *FM-GAN-Tan* (Kumar et al., 2017) | - | 95.61 | 83.80 |
| *Bad GAN* (Dai et al., 2017) | **99.20** | 95.75 | **85.59** |
| *VAT* (Miyato et al., 2017) | 98.64 | 93.17 | 85.13 |
| *CrossEnt* (use all data) | 98.82 | 96.74 | 90.31 |
| *CrossEnt* (use lab. data only) | 79.16 | 88.91 | 67.45 |
| *mod. VAT* | 98.70 | 94.69 | 85.18 |
| *FAT* | 98.89 | **95.94** | 85.31 |

Table 2: Comparison of prediction accuracies with small labeled data and more complex dataset.

| Method | Test acc.(%) | | | |
|---|---|---|---|---|
| | MNIST(20) | SVHN(500) | CIFAR10(1000) | CIFAR100(8000) |
| *FM-GAN* (Salimans et al., 2016) | 83.23 | 81.56 | 78.13 | - |
| *FM-GAN-Tan* (Salimans et al., 2016) | - | 95.13 | **80.48** | - |
| *CrossEnt*(use all data) | 98.82 | 96.74 | 90.31 | 46.72 |
| *CrossEnt*(use lab. data only) | 54.92 | 85.82 | 50.30 | 18.35 |
| *mod. VAT* | 89.92 | 92.59 | 75.43 | 34.76 |
| *FAT* | **96.32** | **95.21** | 75.96 | **35.52** |

We use *Adam* algorithm (Kingma & Ba, 2014) to update the parameters and do not use any data augmentation techniques. The results are summarized in Table 1, which shows that *FAT* achieves the state-of-the-art accuracy for SVHN dataset and competitive accuracies with the state-of-the-art method (i.e. *Bad GAN*) for MNIST and CIFAR10.

We conduct another experiments where the numbers of labeled data are much smaller and consider a more complex dataset CIFAR100 (Krizhevsky & Hinton, 2009), whose results are summarized in Table 2. Note that *FAT* still dominates *mod. VAT* for the all datasets and the margins become larger. While *FAT* achieves the state-of-the-art performances for MNIST and SVHN, its performance degrades much for CIFAR10 compared to the accuracy for the case of 4000 labeled data. Note that the quality of bad samples depends on the quality of the estimated classifier. For CIFAR10 which is relatively more complex than MNIST and SVHN, the quality of the estimated classifier is influenced much to the amount of labeled data and thus the accuracy of *FAT* is more sensitive to the amount of labeled data. However, it is interesting to see that *FAT* is superior to *mod. VAT* for CIFAR10 and CIFAR100, which suggests that bad samples are helpful for complex problems where the quality of bad samples might not be sufficiently good.

The other advantage of *FAT* is its stability with respect to learning phase. With small labeled data, the test accuracies of each epoch tends to fluctuate much for *VAT* and *mod VAT*, while *FAT* provides much more stable results. See Figure 7 in Appendix. This may be partly because 'good' bad samples keep the classifier from fluctuation.

Table 3: Test accuracies of MNIST for various values of $\tau$ and $\delta$. The other two parameters on each case are fixed to the optimal values.

| $\tau$ | 0.001 | 0.01 | 0.1 | 0.2 | $\delta$ | 1 | 2. | 4. | 6. |
|---|---|---|---|---|---|---|---|---|---|
| Test acc. | 98.65 | 98.89 | 98.77 | 98.71 | Test acc. | 89.54 | 98.89 | 98.79 | 98.55 |

Table 4: Test accuracies for diverse objective loss functions.

| Data | MNIST | SVHN | CIFAR10 |
|---|---|---|---|
| Setting | | Test acc.(%) | |
| $L^{\text{true}} + L^{\text{fake}} + L^{\text{VAT}}$ | 98.89 | 95.94 | 85.15 |
| $L^{\text{true}} + L^{\text{fake}}$ | 83.6 | 90.21 | 68.32 |
| $L^{\text{fake}} + L^{\text{VAT}}$ | 98.77 | 95.71 | 85.31 |

## 6.2 Effects of tuning parameters

*FAT* introduces two additional tuning parameters $\tau$ and $\delta$ compared to *VAT*, where $\tau$ is used to determine whether a bad sample is 'good' and $\delta$ is the radius to generate bad samples. We investigate the sensitivities of prediction performances with respect to the changes of the values of $\tau$ and $\delta$ with $\epsilon$ being fixed at 1.5 that is the optimal value. The results are reported in Table 3. Unless $\tau$ is too small or too large, the prediction performances are not changed much. For $\delta$, care should be done. Too small $\delta$, smaller than $\epsilon$, hampers the prediction performance much. Apparently, a similar value of $\delta$ to that of $\epsilon$ (i.e. $\delta = 2$) gives the best result.

## 6.3 Objective function analysis

In this subsection, we analyze the effects of $L^{\text{true}}$ and $L^{\text{VAT}}$ in this section. We do not consider the cross-entropy of the labeled data and $L^{\text{fake}}$ since they are necessary to reflect the idea utilizing bad data. Table 4 compares the performances without one or both of these terms. Note that $L^{\text{VAT}}$ is necessary for superior performances, which is because $L^{\text{VAT}}$ is indispensable to generate 'good' bad samples as explained in Section 5.1. On the other hand, the term $L^{\text{true}}$, which is devised to separate unlabeled data from bad samples, is not always helpful. A possible explanation is that sometimes the term $L^{\text{true}}$ would give an undesirable effect on the misclassified data.

## 6.4 Computational efficiency

We investigate the computational efficiency of our method in view of learning speed and computation time per training epoch. For *Bad GAN*, we did not use *PixelCNN++* on SVHN and CIFAR10 datasets since the pre-trained *PixelCNN++* models are not publicly available. Figure 4 draws the bar plots about the numbers of epochs needed to achieve the prespecified test accuracies. We can clearly see that *FAT* requires much less epochs

We also calculate the ratios of the computing time of each semi-supervised learning algorithm over the computing time of the corresponding supervised learning algorithm for CIFAR10 dataset, whose results are summarized in Table 5. These ratios are almost same for different datasets. The computation time of *FAT* is competitive to *VAT* and *mod. VAT*, and hence we can conclude that *FAT* arrives at the prespecified performances much efficiently. Note that *PixelCNN++* is not used for this experiment, and so comparison of computing time of *FAT* and *Bad GAN* with *PixelCNN++* is meaningless

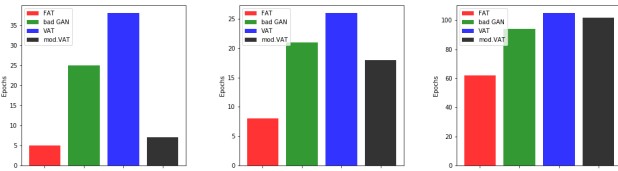

Figure 4: The number of epochs to achieve the prespecified test accuracies (98%, 90% and 80%) with the four methods for **(Left)** MNIST, **(Middle)** SVHN and **(Right)** CIFAR10 datasets. *Bad GAN* is operated without *PixelCNN++* for SVHN and CIFAR10 datasets.

Table 5: Learning time per training epoch ratios compared to supervised learning with cross-entropy for CIFAR10. *Bad GAN* is operated without *PixelCNN++*.

| Method | *VAT* | *mod. VAT* | *FAT* | *Bad GAN* |
|---|---|---|---|---|
| Time ratio | 1.37 | 1.62 | 2.09 | 3.20 |

### 6.5 Quality of bad samples generated by adversarial training

We investigate how 'good' bad samples generated by the adversarial training are. The upper panel of Figure 5 shows the scatter plot of the synthetic data and the trace plot of prediction accuracies of *FAT* and *VAT*. And the lower panel of Figure 5 draws the scatter plots with generated bad samples at various epochs. We can clearly see that bad samples move to lower density regions as the epoch increases, which amply demonstrates that the adversarial training is good at generating bad samples. We also compare bad images of the MNIST data generated by *FAT* and *Bad GAN* in Figure 6. The bad images by *FAT* do not look like real images and do not seem to be collapsed, which indicates that *FAT* consistently generates diverse and good bad samples. *Bad GAN* also generates diverse bad samples but some 'realistic' images can be found.

## 7 Conclusion

In this paper, we propose a new method called *FAT* for semi-supervised learning which generates bad samples only with a given classifier. The objective function of *FAT* is devised to compromise the advantages of *Bad GAN* and *VAT* together, which makes *FAT* be faster and more accurate. In numerical experiments, we show that *FAT* achieves almost the state-of-the-art performances with much fewer epochs. Unlike *Bad GAN* , *FAT* only needs to learn a discriminator. Hence, it could be extended without much effort to other learning problems. For example, *FAT* can be modified easily for recurrent neural networks and hence can be applied to sequential data. We will leave this extension as a future work.

It would be useful to combine *FAT* with a generative approach such as *Bad GAN*, in particular when the initial classifier is bad. Since *FAT* uses only a classifier to generate bad samples, the initial estimate of the classifier would be important. Using the generator model learned by large unlabeled data would be helpful to find a good initial estimate of the classifier.

### Acknowledgments

This work is supported by Samsung Electronics Co., Ltd.

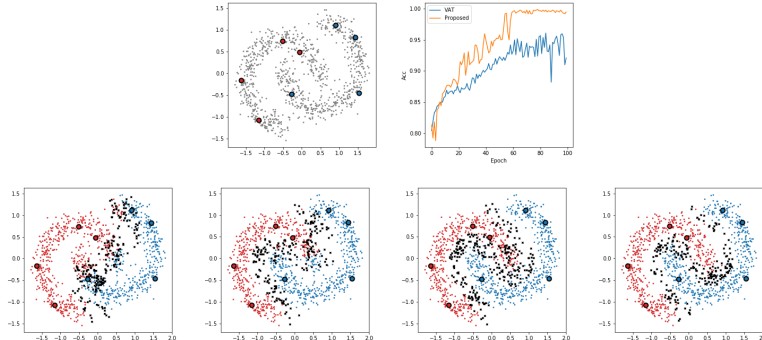

Figure 5: **(Upper Left)** The scatter plot of synthetic data which consist of 1000 unlabeled data (gray) and 4 labeled data for each class (red and blue with black edge). **(Upper Right)** Accuracies of unlabeled data for each epochs for *VAT* and *FAT*. We use 2-layered NN with 100 hidden units each. **(Lower)** Bad samples and classified unlabeled data by colors at the different training epochs of *FAT*

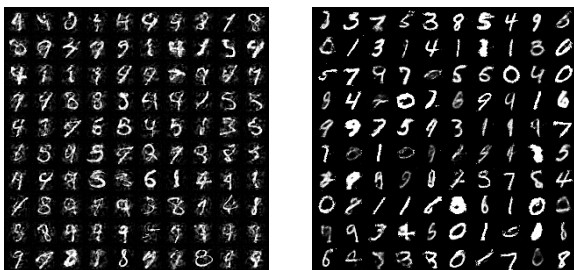

Figure 6: 100 randomly sampled bad images using **(Left)** *FAT* and **(Right)** *Bad GAN*.

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

# A APPENDIX

## A.1 PROOF OF PROPOSITION 1

Without loss of generality, we assume that $w'x + b > 0$, that is, $p(y = 1|x; \eta) > p(y = 0|x; \eta)$. We will show that there exists $\epsilon > 0$ such that $w'r^*(x, \epsilon) < 0$. Note that

$$\operatorname*{argmax}_{r, ||r|| \leq \epsilon, w'r > 0} KL(x, r; \eta) = \epsilon \frac{w}{||w||} (=: r_1^*) \quad \text{and}$$

$$\operatorname*{argmax}_{r, ||r|| \leq \epsilon, w'r < 0} KL(x, r; \eta) = -\epsilon \frac{w}{||w||} (=: r_2^*).$$

So all we have to do is to show

$$KL(x, r_2^*; \eta) > KL(x, r_1^*; \eta).$$

By simple calculation we can get the following:

$$KL(x, r_2^*; \eta) - KL(x, r_1^*; \eta) = - \left[ p(y = 1|x; \theta) w' (r_2^* - r_1^*) - \log \frac{\exp\left(w'(x + r_2^*) + b\right) + 1}{\exp\left(w'(x + r_1^*) + b\right) + 1} \right].$$

Using the Taylor's expansion up to the third-order, we obtain the following:

$$\begin{aligned}
\log\left[\exp\left(w'(x + r) + b\right) + 1\right] &= \log\left[\exp\left(w'x + b\right) + 1\right] + p(y = 1|x; \eta)w'r \\
&\quad + \frac{1}{2} p(y = 1|x; \eta)p(y = 0|x; \eta)r'ww'r \\
&\quad - \frac{1}{6} p(y = 1|x; \eta)p(y = 0|x; \eta) \left\{p(y = 1|x; \eta) - p(y = 0|x; \eta)\right\} \sum_{i,j,k=1}^{p} w_i w_j w_k r_i r_j r_k \\
&\quad + o(||r||^3).
\end{aligned}$$

So,

$$\begin{aligned}
\log \frac{\exp\left(w'(x + r_2^*) + b\right) + 1}{\exp\left(w'(x + r_1^*) + b\right) + 1} &= p(y = 1|x; \eta)w'(r_2^* - r_1^*) \\
&\quad + \frac{1}{3} p(y = 1|x; \eta)p(y = 0|x; \eta)\left\{p(y = 1|x; \eta) - p(y = 0|x; \eta)\right\} \epsilon^3 ||w||^3 + o(\epsilon^3).
\end{aligned}$$

Thus, we have the following equations:

$$\begin{aligned}
KL(x, r_2^*; \eta) - KL(x, r_1^*; \eta) &= \frac{1}{3} p(y = 1|x; \eta)p(y = 0|x; \eta)\left\{p(y = 1|x; \eta) - p(y = 0|x; \eta)\right\} \epsilon^3 ||w||^3 + o(\epsilon^3) \\
&= C \cdot \epsilon^3 + o(\epsilon^3).
\end{aligned}$$

Therefore, there exists $\epsilon^* > 0$ such that $KL(x, r_2^*; \eta) > KL(x, r_1^*; \eta)$ for $\forall 0 < \epsilon < \epsilon^*$. $\square$

## A.2 EXTENSION OF PROPOSITION 1 FOR THE DNN CLASSIFIER

Consider a binary classification DNN model with ReLU activation function $p(y = 1|x; \theta) = (1 + \exp(-g(x; \theta)))^{-1}$ parameterized by $\theta$. Since $g(x; \theta)$ is piecewise linear, we can write $g(\cdot; \theta)$ as

$$g(\cdot; \theta) = \sum_{j=1}^{N} \mathbb{I}(\cdot \in \mathcal{A}_j) \cdot (w_j x + b_j),$$

where $\mathcal{A}_j$ is a linear region and $N$ is the number of linear regions.

For given $x$, suppose $g(x;\theta) > 0$. If $g(x:\theta)$ is estimated reasonably, we expect that $g(x;\theta)$ is decreasing if $x$ moves toward the decision boundary. A formal statement of this expectation would be that $x - r\nabla_x g(x;\theta)$ can arrive at the decision boundary for a finite value of $r > 0$, where $\nabla_x$ is the gradient with respect to $x$. Of course, for $x$ with $g(x;\theta) < 0$, we expect that $x + r\nabla_x g(x;\theta)$ can arrive at the decision boundary for a finite value of $r > 0$. We say that $x$ is *normal* if there is $r > 0$ such that $x - r\nabla_x g(x:\theta)\text{sign}\{g(x:\theta)\}$ locates at the decision boundary. We say that a linear region $\mathcal{A}_j$ is *normal* if all $x$ in $\mathcal{A}_j$ are *normal*. We expect that most of $\mathcal{A}_j$ are *normal* if $g(x;\theta)$ is reasonably estimated so that the probability decreases or increases depending on $\text{sign}\{g(x:\theta)\}$ if $x$ is getting closer to the decision boundary.

The following proposition proves that the adversarial direction is toward the decision boundary for all $x$s in *normal* linear regions.

**Proposition 3** *If a linear region $\mathcal{A}_j$ is normal. Then for any $x \in int(\mathcal{A}_j)$, there exists $\epsilon > 0$ and $\delta > 0$ such that $x_{bad} = x + \delta r_{advr}(x,\epsilon)/||r_{advr}(x,\epsilon)||$ is on the decision boundary.*

*Proof.* Take $\tilde{\epsilon} > 0$ such that $x + r \in \mathcal{A}_{\tilde{j}}$ for $\forall r \in B(x,\tilde{\epsilon})$. Then by Proposition 1, there exists $0 < \epsilon^* < \tilde{\epsilon}$ such that for $\forall 0 < \epsilon < \epsilon^*$,

$$
\begin{aligned}
r_{\text{advr}}(x,\epsilon) &= \epsilon \cdot \text{sign}(-b_{\tilde{j}} - w_{\tilde{j}}'x)) \cdot \frac{w_{\tilde{j}}}{||w_{\tilde{j}}||} \\
&\propto -\nabla_x g(x;\theta)\text{sign}\{g(x:\theta)\}.
\end{aligned}
$$

$x$ is *normal*, thus there exists $\delta > 0$ such that $x_{\text{bad}} = x + \delta r_{\text{advr}}(x,\epsilon)/||r_{\text{advr}}(x,\epsilon)||$ belongs to the decision boundary. $\square$

### A.3 PROOF OF PROPOSITION 2

It suffices to show that $\forall x, x' \in \mathcal{U}_k^{tr}, f(x) = f(x')$ for all $k = 1, ..., K$. For given $\mathcal{U}_k^{tr}$, there exists $(\tilde{x}, \tilde{y}) \in \mathcal{L}^{tr}$ such that $d(\{\tilde{x}\}, \mathcal{U}_k^{tr}) < \epsilon$. So $\mathcal{U}_k^{tr} \cup \{\tilde{x}\}$ is $\epsilon$-*connected*. That is, for any $x \in \mathcal{U}_k^{tr}$, there exists a path $(\tilde{x}, x_1, ..., x_q, x)$ such that $(\tilde{x}, x_1), ..., (x_q, x)$ are all $\epsilon$-*connected*. Therefore $y(x) = y(\tilde{x}) = \tilde{y}$ for $\forall x \in \mathcal{U}_k^{tr}$, and the proof is done. $\square$

A.4   FIGURE

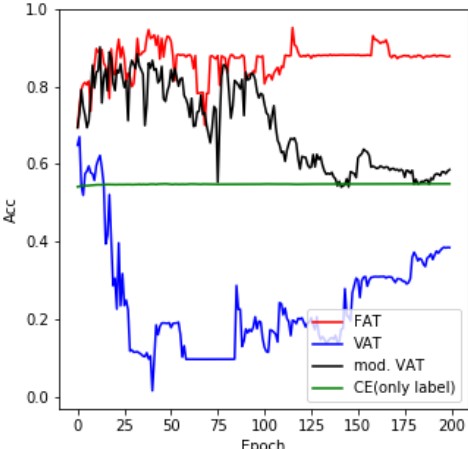

Figure 7: Trace plot of the test accuracies with the four methods for MNIST dataset. 20 randomly sampled data are used as the labeled data and the rest are used as the unlabeled data.

