# OpenReview forum: "Fast adversarial training for semi-supervised learning"
_ICLR.cc/2019/Conference_

### Official Review · AnonReviewer3 · 2018-10-31

**Rating:** 5
**Confidence:** 4

**Review:**

The paper proposes to use the technique in VAT to generate adversarial complementary examples in the (K+1)-class semi-supervised learning framework described by the Bad GAN paper. This leads to a formulation that combines the VAT loss and the (K+1)-class classification loss. The paper also provides analysis regarding why VAT is useful for semi-supervised learning.

Pros
1. It is interesting to bridge two state-of-the-art semi-supervise learning methods in a meaningful.
2. Some positive results have been presented in Table 1 and Figure 4.

Cons and questions
1. I don't understand the authors' claim that FAT uses both pushing and pulling operations. It might be true that both Bad GAN and VAT encourage a decision boundary in the low-density region, but how are they different? Are pushing and pulling really different things here?
2. Unfortunately the proposed method does not give substantial improvement over Bad GAN or VAT in terms of accuracy.
3. If using VAT to generate bad samples is a reasonable approach, then based on the theory in Dai et al., the Bad GAN formulation would not need the additional VAT regularization term to guarantee generalization. On the other hand, based on the theory of Proposition 2, VAT itself should be sufficient. Why do we still need the (K+1)-class formulation. It seems that combination of Bad GAN and VAT objectives has not been well motivated or fully justified. Does this explain the fact that not much empirical gain was obtained by this method?
4. The authors try to use Proposition 1 to motivate the use of VAT for generating complementary examples. However, it seems that the authors misinterprets the concept of bad examples proposed in Dai et al. The original definition (which led to the theoretical guarantees in Dai et al) of bad examples is low-density data samples. In the current paper, the authors assume that data samples close to decision boundaries are bad examples. This is not sound because low-density samples are not equivalent to samples close to decision boundaries, especially when the classifier is less perfect. As a result, the theoretical justification of using VAT to sample complementary examples is a bit weak.
5. There is not ablation study of different terms in the objective function.
6. In Figure 4, you can compare your method with Bad GAN without a PixelCNN. Bad GAN does not need a PixelCNN to achieve the reported results in their paper, and their results are reproducible by running the commands given in the github repo. It would be good to add this comparison.

---

> ### Author Response · Authors · 2018-11-26
> **Response to reviewer #3**
>
> Response to reviewer 3.
>
> We thank you for your thoughtful review and comments. We give a response to each of your comments in cons and questions:
>
> - I don't understand the authors' claim that FAT uses both pushing and pulling operations. It might be true that both Bad GAN and VAT encourage a decision boundary in the low-density region, but how are they different? Are pushing and pulling really different things here?
>
> A. As you addressed, both Bad GAN and VAT encourage a decision boundary in the low-density region. Bad samples are assumed to locate at low density regions and they pull the decision boundary to the low density regions where they locate. In contrast, the VAT encourages the class probabilities of the true data not changing much. This is achieved if the decision boundary does not locate at the areas where the true data live. In this sense, we say that the VAT push the decision boundary from the high density regions.
>
> - Unfortunately the proposed method does not give substantial improvement over Bad GAN or VAT in terms of accuracy.
>
> A. FAT dominates vanilla VAT for all datasets we considered even if the margins are not super large. But, the margins of Bad GAN, which is known to be the state-of-art algorithm, are also not that super large. Considering the margins of Bad GAN with other competitors, we believe that the margins of FAT compared to vanilla VAT are significant for FAT to be a useful method. Also, we investigated the performances with fewer labeled samples and we found that the performance differences between FAT and vanilla VAT become larger, which is stated in Section 6.1. Regarding Bad GAN, FAT does not dominate Bad GAN but note that Bad GAN requires additional very computationally heavy algorithms such as PixcelCNN++ which is not even publicly unavailable. There is a Bad GAN algorithm in github which does not use PixcelCNN++. Bad GAN without PixcelCNN++ is almost the same as FM-GAN which is significantly inferior to FAT for SVHN and CIFAR10.
>
> - If using VAT to generate bad samples is a reasonable approach, then based on the theory in Dai et al., the Bad GAN formulation would not need the additional VAT regularization term to guarantee generalization. On the other hand, based on the theory of Proposition 2, VAT itself should be sufficient. Why do we still need the (K+1)-class formulation. It seems that combination of Bad GAN and VAT objectives has not been well motivated or fully justified. Does this explain the fact that not much empirical gain was obtained by this method?
>
> A. The main idea of FAT is to simplify Bad GAN by generating bad samples only by use of the current estimated classifier without generator. But, for this idea to work well, the current classifier should be reasonably well and VAT does something for this as explained by 5.1. On the other hand, we may think FAT as an improved version of VAT by adding Bad GAN idea. Theoretically, VAT can find a good classifier but it frequently happens that theoretical results are not realized in practice. Our contribution is that combining the idea of Bad GAN with VAT, we can realize the advantage of VAT more easily in real data.

---

> > ### Author Response · Authors · 2018-11-26
> > **Response to reviewer #3 (cont.)**
> >
> > - The authors try to use Proposition 1 to motivate the use of VAT for generating complementary examples. However, it seems that the authors misinterprets the concept of bad examples proposed in Dai et al. The original definition (which led to the theoretical guarantees in Dai et al) of bad examples is low-density data samples. In the current paper, the authors assume that data samples close to decision boundaries are bad examples. This is not sound because low-density samples are not equivalent to samples close to decision boundaries, especially when the classifier is less perfect. As a result, the theoretical justification of using VAT to sample complementary examples is a bit weak.
> >
> > A. We agreed. Bad samples are not necessarily close to the decision boundary. However, in the conditions of Dai et al. (2017), they assumed that bad samples should locate in between true data classes where no true data exist. Hence, there is no problem when low density regions and the decision boundary coincide. A possible problem of FAT is that the estimated classifier may not locate at the low density region. In this case, generating bad samples using the estimated classifier may not work well. We know this problem and we agree that FAT only works when the initial classifier is reasonable. However, note that VAT term in FAT plays a key role to find a reasonable initial classifier as explained in Section 5 of the revised manuscript.
> >
> > - There is not ablation study of different terms in the objective function.
> >
> > A. We did ablation study about the various terms in the objective function of FAT in Section 6.3. We found that L^VAT is indispensable but the role of L^true is somehow mixture.
> >
> > - In Figure 4, you can compare your method with Bad GAN without a PixelCNN. Bad GAN does not need a PixelCNN to achieve the reported results in their paper, and their results are reproducible by running the commands given in the github repo. It would be good to add this comparison.
> >
> > A. Thank you for your suggestion. As you recommended, we conducted Bad GAN for SVHN and CIFAR10 datasets by running the codes given in the github and added the results in Figure 4. The results again confirm that FAT achieves competitive results with much fewer training epoch and training time than other existing researches. In addition, the test accuracies of Bad GAN without PixcelCNN++ are significantly inferior to FAT (the accuracies of Bad GAN without PixcelCNN++ are similar to those of FM-GAN) even though we did not report them in the revised paper.

---

### Official Review · AnonReviewer2 · 2018-11-02
**Some interesting observations, but not quite convincing just yet**

**Rating:** 5
**Confidence:** 4

**Review:**

This paper makes the interesting observation that the generative procedure proposed by Bad GAN paper can be replaced by a slightly modified VAT procedure. The reasoning is sound and leverages the intuition that adversarial examples (subject to a sufficiently small perturbation radius) are likely to be closer to a decision boundary than the original sample.

The paper is generally easy to follow but the presentation could be improved. In particular more could be done to describe the terms in Equation 5. I’m also curious about the behavior of L^true, which is equivalently the fourth term in Eq 1. Even when reading Bad GAN paper, I did not quite understand their claim that this can be correctly interpreted as a conditional entropy term (if they really wanted conditional entropy, they should probably have either done H(p(k|x)) or H(p(k|x, k <= K))). I agree with the authors that the roles of the second and fourth terms overlapped, and I think this is sufficiently interesting to warrant some further elaboration in the paper. I also liked the reminder that power iteration selects a non-unique sign for the first eigenvector (subject to the random vector initialization); I encourage the authors to do an ablation test to convince the reader that “this modification helps to improve convergence speed of the test accuracy.”

The propositions in this paper were, in my opinion, not particularly insightful. While I think it is nice that the authors went through the effort of providing some formalism to the intuition that VAT has a “push decision boundary away from high-density regions”, I’m less sure if propositions 1 and 2 really provides any additional insight the behavior of VAT. Proposition 1 is pretty weak in that it only covers a 2-class logistic regression; it seems obvious that the adversarial perturbation points in the direction toward the decision hyperplane. If the authors could extend this to more general non-linear classifiers (perhaps subject to some assumptions), that would be more interesting. I don’t think Proposition 2 has any real value and recommend its relegation to the appendix.

I think the biggest weakness of this paper is the experiments. Taking Table 1 at face value, the conclusion that FAT is simply competitive with existing approaches suggests that the additional machinery isn’t particularly useful, providing little more than a vanilla VAT. I also think MNIST/SVHN has run its course as good semi-supervised learning benchmarks and would prefer to see such algorithms being scaled to more complex data. The main argument for why FAT should be prefered over VAT comes from Section 6.2. Figure 4 is more interesting, but is complicated by the fact that FAT checks both possible eigenvectors (+/- u) during training, which requires two forward passes in the classifier; did the authors give a similar treatment to VAT? Please show wall-clock time too. Unfortunately the computational efficiency gain seems to only hold true for MNIST/SVHN, but not for CIFAR. I worry that the observed gains will not sustain once we move to more complicated datasets.


Pros:
+ Simple and clean proposal
+ Easy to read
Cons:
- Limited insight
- Weak experiments

---

> ### Author Response · Authors · 2018-11-26
> **Response to reviewer #2**
>
> Response to reviewer 2.
>
> Thank you for your thoughtful review and comments. We give a response to each of your comments in cons and questions:
>
> - In particular more could be done to describe the terms in Equation 5. I’m also curious about the behavior of L^true, which is equivalently the fourth term in Eq 1. Even when reading Bad GAN paper, I did not quite understand their claim that this can be correctly interpreted as a conditional entropy term (if they really wanted conditional entropy, they should probably have either done H(p(k|x)) or H(p(k|x, k <= K))).
>
> A. Yes, L^true is not a conditional entropy. But utilizing L^true is a more reasonable choice than H(p(k|x)) or H(p(k|x,k<=K)).
> Dai et al. (2017) stated the following three sufficient conditions for Bad GAN to find a perfect classifier:
> i) A classifier classifies the labeled data perfectly, that is, argmax_k g_k(x;\theta)=y for all (x,y) in labeled data.
> ii) Unlabeled data are classified to real label, not bad label, that is, max_k g_k(x;\theta)>0 for all x in unlabeled data.
> iii) Bad data are classified to bad label, that is, max_k g_k(x;\theta)<0 for all x in bad data.
>
> L^true is introduced to encourage the classifier to satisfy the condition ii). Our understanding about L^true is as follows: Maximizing H(p(k|x)) or H(p(k|x,k<=K)) does not guarantee the condition ii). Firstly suppose that we use H(p(k|x)) and there exists a real datum x such that the current classifier classifies x to the bad label, i.e. max_k g_k(x;\theta)<0. Then minimizing H(p(k|x)) encourages max_k g_k(x;\theta) to become negatively smaller which is undesirable for semi-supervised learning. On the other hand, using H(p(k|x,k<=K)) is also problematic since minimizing H(p(k|x,k<=K)) is nothing to do with max_k g_k(x;\theta)>0 because H(p(k|x,k<=K)) just maximizes the relative differences between g_k(x;\theta)s and hence there may exist a real datum x with large H(p(k|x,k<=K)) but negative max_k g_k(x;\theta). Thus, L^true is more helpful than H(p(k|x)) or H(p(k|x,k<=K)) for Bad GAN.
>
> - I encourage the authors to do an ablation test to convince the reader that “this modification helps to improve convergence speed of the test accuracy.”
>
> A. We investigated the effects of the modification and added these results in Table 1 and Figure 4. We can clearly see that the modified VAT does improve the test accuracy and the convergence speed but is inferior to FAT.
>
> - If the authors could extend this to more general non-linear classifiers (perhaps subject to some assumptions), that would be more interesting.
>
> A. We added a discussion about the relation of adversarial direction and bad samples for the DNN with ReLU activation function in Appendix.
>
> - I don’t think Proposition 2 has any real value and recommend its relegation to the appendix.
>
> A. One of the main message of FAT is that the role of VAT is different from the role of bad samples. We believe that explanation of the role of VAT is a key ingredient of the proposed method and hence we decided to keep it in the main body. But we condensed it as much as possible.

---

> > ### Author Response · Authors · 2018-11-26
> > **Response to reviewer #2 (cont.)**
> >
> > - I think the biggest weakness of this paper is the experiments. Taking Table 1 at face value, the conclusion that FAT is simply competitive with existing approaches suggests that the additional machinery isn’t particularly useful, providing little more than a vanilla VAT.
> >
> > A. FAT dominates vanilla VAT for all datasets we considered even if the margins are not super large. But, the margins of Bad GAN, which is known to be the state-of-the-art algorithm, are also not that super large. Considering the margins of Bad GAN with other competitors, we believe that the margins of FAT compared to vanilla VAT are significant for FAT to be a useful method. Also, we investigated the performances with fewer labeled samples and we found that the performance differences between FAT and vanilla VAT become larger.
> >
> > - I also think MNIST/SVHN has run its course as good semi-supervised learning benchmarks and would prefer to see such algorithms being scaled to more complex data.
> >
> > A. Thank you for your recommendation. We used the three well-known datasets first because VAT and bad GAN used these datasets. For FAT combines VAT and bad GAN, we thought that the comparisons of FAT with VAT and bad GAN over these datasets should be preceded. However, as you recommended, we applied FAT and modified VAT to CIFAR 100 and we found that FAT outperformed modified VAT, which is summarized in Section 6.1. Note that CIFAR 100 is very difficult to be classified and hence bad samples generated by the estimated classifier may not be ‘good’ bad samples. Based on this argument, we can conclude that FAT is not affected much from ‘not good’ bad samples. To sum up, FAT improves VAT and FAT is robust to ‘not good’ bad samples.
> >
> > - The main argument for why FAT should be prefered over VAT comes from Section 6.2. Figure 4 is more interesting, but is complicated by the fact that FAT checks both possible eigenvectors (+/- u) during training, which requires two forward passes in the classifier; did the authors give a similar treatment to VAT? Please show wall-clock time too.
> >
> > A. In the revised manuscript, we added the results of the modified VAT. We found that the modified VAT improves the vanilla VAT but still is dominated by FAT in the three benchmark datasets. In addition, we added results of computing times in Section 6.4 of the revised manuscript.
> >
> > - Unfortunately the computational efficiency gain seems to only hold true for MNIST/SVHN, but not for CIFAR. I worry that the observed gains will not sustain once we move to more complicated datasets.
> >
> > A. We redid comparison of computational complexity (based on the numbers of epochs to arrive at the prespecified test accuracies) and found that FAT is more efficient than the other competitors for not only MNIST and SVHN but also CIFAR10. See Section 6.4 for the results. Apparently, there are two reasons for these differences. First, in the initial experiment, we did not know some randomness of PyTorch besides the manual seed to control GPU and thus the results in the original manuscript would not be a fair comparison. In the revised manuscript, we fixed the randomness in Pytorch and did several analyses and averaged out the numbers of epochs. The second reason is that we found better hyperparameters for CIFAR10. Note that the accuracy of CIFAR10 in the revised manuscript is little bit better than that in the original manuscript (85.31 vs. 85.22).

---

### Official Review · AnonReviewer1 · 2018-11-02
**A good paper on combining BadGAN and VAT for SSL**

**Rating:** 7
**Confidence:** 4

**Review:**

The authors propose to combine BadGAN framework and VAT to accelerate learning in the semi-supervised setting. The paper shows that the VAT approach is actively pushing the decision boundary away from the high-density regions. While the BadGAN approach pulls the decision boundary to low-density regions. This simultaneous push and pull lead to after convergence in testing accuracy. The authors also report competitive results on standard datasets used for SSL such as SVHN and CIFAR10.

Positives:
The approach overcomes some of the difficulties with BadGAN which arise from training a GAN and density estimation network for generating “bad samples” useful for SSL. Instead of using a GAN, the proposed approach uses adversarial samples using VAT that are sufficiently confusing to the current estimate of the classifier.
The theoretical justifications for the VAT interpretation are interesting and convincing. The visualizations of the bad samples show qualitatively that the bad samples from the BadGAN and proposed approach differ. Several other visualization aids in understanding the behavior of the algorithm.

Negatives:
Requires additional hyperparmeter tuning in tau and rho. Tuning these with large validation sets can lead to an overoptimistic estimate of the generalization. How sensitive is the performance to these parameters?
As the authors point out, the method has limitations when the number of labeled samples is much smaller. It will be nice to see some results in this aspect.
Please include more details to clarify what is meant by ‘the role of the second term of (1) and (2) are overlapped’.

---

> ### Author Response · Authors · 2018-11-26
> **Response to Review #1**
>
> Response to reviewer 1.
>
> Thank you for your valuable review and for highlighting the paper is well written. We give a response to each of your comments in cons and questions:
>
> - Requires additional hyperparmeter tuning in tau and rho. Tuning these with large validation sets can lead to an overoptimistic estimate of the generalization. How sensitive is the performance to these parameters?
>
> A. We did sensitivity studies with MNIST data in Section 6.2 in the revised manuscript. We found that the results are quite robust to the choice of \tau unless \tau is too small or too large. For \delta, the performances degrade much when \delta is too small, i.e. smaller than \epsilon, but we found that the performance is similar when \delta is close to \epsilon.
>
> - As the authors point out, the method has limitations when the number of labeled samples is much smaller. It will be nice to see some results in this aspect.
>
> A. We did additional experiment with very few labeled data. Surprisingly, FAT performances better compared to the other competitors for MNIST and SVHN datasets. We added the results of the experiment in Section 6.1.
>
> - Please include more details to clarify what is meant by ‘the role of the second term of (1) and (2) are overlapped’.
>
> A. We changed the statement by “We delete the second term of (1) because it can be easily shown that a perfect classifier of labeled data can be obtained from the minimizer of the objective function (1) without the second term under the conditions in Dai et al. (2017).”

---

### Meta-Review · Area_Chair1 · 2018-12-15
**Potentially interesting idea but motivation remains somewhat unclear; marginal improvements in SSL performance**

**Confidence:** 5
**Recommendation:** Reject

**Metareview:**

The paper combines the ideas of VAT and Bad GAN, replacing the fake samples in Bad GAN objective with VAT generated samples. The motivation behind using the K+1 SSL framework with VAT examples remains unclear, particularly in the light of Prop. 2 which shows smoothness of classifier around the unlabeled examples is enough (which VAT already encourages). R2 and R3 have raised the point of limited insight and lack of motivation behind combining VAT and Bad GAN objectives in this way. R2 and R3 are also concerned about the empirical results which show only marginal improvements over VAT/BadGAN in most settings.

AC feels that the idea of the paper is interesting but agrees with R2/R3 that the proposed objective is not motivated well enough (what is the precise advantage of using K+1 SSL formulation with VAT examples?). The paper really falls on the borderline and could be improved if this point is addressed convincingly.